# Optimality and sub-optimality in a bacterial growth law

Benjamin D. Towbin[1], Yael Korem[1], Anat Bren[1], Shany Doron[2], Rotem Sorek[2] & Uri Alon[1]

Organisms adjust their gene expression to improve fitness in diverse environments. But finding the optimal expression in each environment presents a challenge. We ask how good cells are at finding such optima by studying the control of carbon catabolism genes in *Escherichia coli*. Bacteria show a growth law: growth rate on different carbon sources declines linearly with the steady-state expression of carbon catabolic genes. We experimentally modulate gene expression to ask if this growth law always maximizes growth rate, as has been suggested by theory. We find that the growth law is optimal in many conditions, including a range of perturbations to lactose uptake, but provides sub-optimal growth on several other carbon sources. Combining theory and experiment, we genetically re-engineer *E. coli* to make sub-optimal conditions into optimal ones and vice versa. We conclude that the carbon growth law is not always optimal, but represents a practical heuristic that often works but sometimes fails.

[1] Department of Molecular Cell Biology, Weizmann Institute of Science, Rehovot 76100, Israel. [2] Department of Molecular Genetics, Weizmann Institute of Science, Rehovot 76100, Israel. Correspondence and requests for materials should be addressed to U.A. (email: urialon@weizmann.ac.il).

To maximize their fitness, organisms need to make appropriate choices to best use their limited resources. But organisms face diverse environments, and in each environment the optimal resource allocation is different. This raises the challenge of finding the optimal response in the large number of possible environments that organisms encounter. Studies of decision making in humans and animals reveal that they make heuristic calculations, known as rules of thumb, that often work but sometimes fail[1–3]. Cells also face different environments and need to allocate their resources appropriately[4,5]. We ask whether cells also use practical heuristics, or—as is often assumed in analysis of cell circuits[4,6,7]—evolve accurate regulatory mechanisms that allow them to be optimal under all conditions. To explore this question, we use resource allocation in the bacterium *E. coli* as a model system.

*E. coli* partitions its resources according to simple linear rules as a function of growth rate, called growth laws[4,8,9]. The expression of most proteins for biomass synthesis, such as ribosomes, increases linearly with growth rate[8]. Conversely, the expression of most enzymes for nutrient uptake and catabolism decreases approximately linearly with growth rate[4]. At least two explanations are possible for these growth laws. First, the laws can represent the optimal solutions, as suggested by several elegant models describing cellular resource allocation[4–7,10–12]. One prediction from this picture is that the growth rate is optimal under all conditions that respect the growth laws and that sub-optimal resource allocation only occurs, when cells deviate from these laws. Previous studies identified conditions of sub-optimal growth in *E. coli*, but did not address this sub-optimality in the context of growth laws[13–15] or focused specifically on conditions that do not follow the growth law[16], so that it remains unclear whether growth laws maximize the growth rate. A second possible explanation for growth laws is that they are practical heuristics: the growth law line is determined by cellular regulation and not by optimality constraints. In this scenario, regulation is expected to maximize the growth rate in many conditions, but to lead to sub-optimal growth in other conditions.

Here we experimentally test the optimality of the carbon growth law—the linear relation between growth rate and resource allocation to carbon catabolism. We find that the carbon growth law provides optimal resource allocation under many conditions, including a wide range of perturbations to lactose uptake. However, on several other carbon sources the growth rate can be improved by experimentally forcing cells to break the growth law. We conclude that linear growth laws are not always optimal for rapid growth. We suggest that growth laws emerge from a transcriptional feedback mechanism that encodes optimal gene control under some conditions, but that is sub-optimal for rapid growth under other conditions.

## Results

**Open-loop control tests optimality of the carbon growth law.** To test the optimality of a growth law, we chose the well-studied carbon catabolism system, controlled by the regulatory molecule cyclic AMP (cAMP). cAMP activates the transcription factor CRP, which controls the expression of hundreds of proteins, including many carbon catabolic enzymes[17] and is also involved in coordinating nitrogen and carbon metabolism[4,18]. High internal carbon concentrations negatively affect cAMP concentrations forming a negative feedback circuit[4,19] (Fig. 1a). We determined growth rate and the activity of CRP (denoted CRP*) using a green fluorescent protein (GFP) reporter at high temporal resolution (9 min) throughout exponential growth in a robotic multi-well plate reader[20] with day–day errors of 4% for growth

rate and 12% for CRP* (Methods; Supplementary Fig. 1). Measurements on 12 different carbon sources (using saturating nitrogen concentration of 18.5 mM $NH_4Cl$) confirmed the growth law of wild-type *E. coli*—an approximately linear relation between allocation to carbon catabolism (CRP*) and growth rate (Fig. 1c; Supplementary Fig. 2a). This linear growth law is also called the C-line[4], where C stands for carbon, and in the present context also for closed-loop control.

To evaluate whether the C-line maximizes the growth rate, we build on previous work employing titration of metabolic gene expression[21–23]. We broke the feedback loop that determines the C-line by adding exogenous cAMP to a strain that cannot endogenously produce it (a strain deleted for the enzymes *cyaA* and *cpdA* that synthesize and degrade cAMP[24]). This design creates an open-loop system, where we modulated CRP* by externally supplying cAMP and measured the resulting growth rate in a given carbon source (Fig. 1b). We call this relation the O-curve, where O stands for open loop. Optimal control means that the endogenous growth rate matches the maximum of the O-curve.

**The growth law maximizes the growth rate on lactose.** We begin with studying growth on lactose, probably the best understood carbon system[25]. The O-curve for growth on lactose is inverse-U shaped, with a maximum at intermediate CRP* (Fig. 1d). This inverse-U shape is due to growth limitation by carbon uptake at low CRP* and due to growth limitation by lack of ribosomes[26–28] and enzyme toxicity[29] at high CRP*. Importantly, the maximum of the O-curve was within 3% of the endogenous growth rate and close to the endogenous control point on the C-line (Fig. 1c,d; Supplementary Table 1, O-curve maximum and s.e. evaluated by parabolic fit to 3 day–day repeats). We conclude that the growth law is optimal for lactose within the precision of the measurements.

To see how robust the computation of optimal resource allocation by the cells is, we perturbed lactose uptake by adding various amounts of a competitive inhibitor of lactose import—the lactose permease LacY inhibitor thio-di-glucoside. The inhibitor reduced the growth rate, and CRP* increased proportionally in accordance with the C-line (Fig. 1e). Importantly, the O-curves also shifted so that their maxima corresponded with the C-line and with the endogenous control (Fig. 1e–g). This correspondence shows that cells responded nearly optimally to the carbon limitation caused by the LacY inhibitor. Similarly, over-expression of lactose metabolic genes by deletion of the lac repressor (*lacI*) was matched by a shift of the O-curve along the C-line that yielded nearly optimal growth response (Supplementary Fig. 2b–d). We conclude that over a large range of lactose uptake rates and perturbations, *E. coli* follows the growth law and tunes its carbon catabolism allocation to achieve the fastest possible growth rate. Such control makes *E. coli* much less sensitive to inhibition of lactose uptake than strains where the control of CRP* is abolished (as assayed using a cAMP-independent CRP mutant in Supplementary Fig. 3).

**The growth law is not optimal on several carbon sources.** To test whether the growth law always maximizes the growth rate, we measured the O-curves for seven additional carbon sources, all of which obeyed the growth law (Fig. 1c; Supplementary Fig. 2a). The O-curves indicate nearly optimal control for four of these carbon sources: maltose, arabinose, sorbitol and glucose (O-curve maximum matches endogenous growth rate to better than 5%; Fig. 2a; Supplementary Table 1).

However, the O-curves indicated that *E. coli* shows non-optimal growth rate on pyruvate, glycerol or galactose (Fig. 2b),

such that the growth rate could be improved by 20–100% by reducing CRP* (Fig. 2b; Supplementary Table 1). In the case of glycerol, our findings are consistent with evolutionary experiments that show that *E. coli* can rapidly evolve on glycerol to reach a faster growth rate with reduced cAMP concentrations[13,15,30].

We conclude that the C-line is not the union of all resource distributions maximizing the growth rate. Instead, the C-line appears to be optimal for growth in many conditions, but sub-optimal for growth in other conditions.

**A mathematical model predicts when the growth law is optimal.** We next sought to understand what makes the growth law fail or succeed in a given condition. To address this question, we analyse a simple model for carbon resource allocation. We then experimentally test the model predictions on optimality by engineering circuits that break or restore optimal control.

We build on detailed modelling of the cAMP system (reviewed in ref. 31) to arrive at a minimal model that is analytically solvable. Carbon catabolites are represented by the variable $x$ that stands for the precursors for biomass synthesis (Fig. 3a; Supplementary Note 1 for complete model description). The growth rate $\mu$ is proportional to the rate of biomass synthesis carried out by ribosomes $R$, with Michaelis–Menten dependence on $x$: $\mu = \gamma R \frac{x}{x + k_2}$. At steady state ($\dot{x} = 0$) the import rate of $x$ is equal to the removal rate of $x$ by biomass production, such that:

$$\beta P(C) \frac{k_1}{k_1 + x} = \gamma R \frac{x}{x + k_2},$$

with carbon import rate $\beta$ that depends on the carbon source, which is reduced by allosteric inhibition of pumps[32] with half-way point $k_1$. The expression of the enzymes for uptake of the carbon source is a function $P(C)$ of the carbon sector $C$, where $C$ is defined as the fraction of the proteome regulated by CRP. The level of most uptake enzymes is proportional to $C$ (refs 33–35), such that $P(C) = C$. Finally, since total protein concentration is approximately constant, we can normalize out all other proteins whose concentration does not change under the present conditions, and set $R + C = 1$ (Supplementary Note 1)[4,27].

The model allows calculation of O-curves as the steady-state growth rate $\mu$ as a function of $C$ (Supplementary Note 2). The modelled O-curves fit the measured O-curves well ($R^2 = 0.94$; Fig. 1g; Supplementary Fig. 4a).

Cells determine the size of the C-sector using a feedback loop in which intracellular carbon $x$ inhibits cAMP[4,19]. We describe this by a Michaelis–Menten repression function between 0 and 1, where $C = 1$ means that all cellular resources go to carbon uptake:

$$C = f(x) = \frac{k_f}{k_f + x}$$

The regulatory constraint given by $f(x)$ defines the modelled C-line (Fig. 3b; Supplementary Note 4). We find that the modelled C-line, which resembles a straight line with some

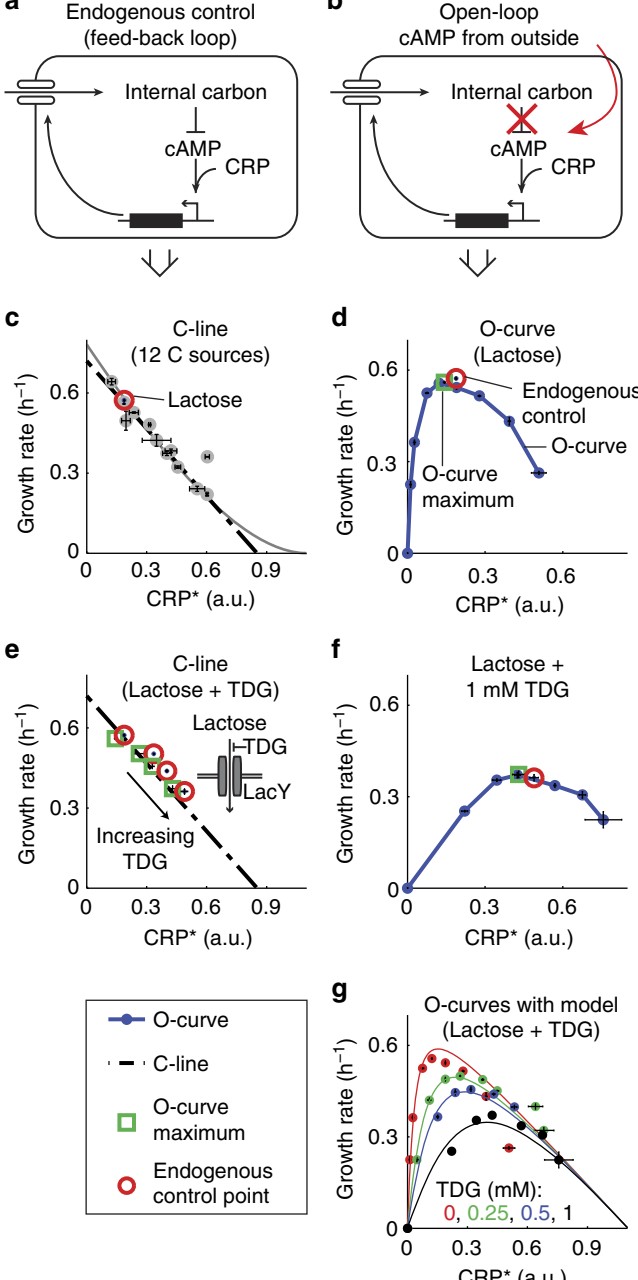

**Figure 1 | Carbon growth law maximizes the growth rate in lactose.**
(**a**) Endogenous control of CRP activity employs a negative feedback loop. Carbon catabolic enzymes, such as carbon pumps, are controlled by the transcription factor CRP, which is activated by the signalling molecule cAMP. cAMP synthesis is repressed by internal carbon. (**b**) To evaluate optimality, we employ an open-loop control of CRP activity. A $\Delta cyaA \, \Delta cpdA$ mutant is used to break feedback control on cAMP signalling, so that CRP activity (denoted CRP*) can be modulated by adding different concentrations of exogenous cAMP to the medium. (**c**) Growth rate and CRP activity of wild-type *E. coli* on 12 different carbon sources decreases linearly with CRP*, defining the C-line. The red circle marks lactose. Black dotted line: best fit line, grey line: model (Fig. 3). Ribose deviates from the line, possibly due to a role of ribose in control of nucleic acid synthesis, and this point was excluded for fits of the C-line (Supplementary Note 5). (**d**) The O-curve is the relation between growth and CRP activity in the open-loop system. The O-curve on lactose shows a maximum that matches the values shown by the endogenous circuit. Green square: O-curve maximum, interpolated from a parabolic fit to the measurement points flanking the point with maximal growth rate. Red circle: endogenous control point (the growth rate and CRP activity of the wild-type strain on the C-line, as in **c**). (**e**) Endogenous system on lactose stays on the C-line even when perturbed by the competitive lactose permease inhibitor thio-di-glucoside (TDG). TDG concentrations were 0, 0.25, 0.5 and 1 mM. (**f**) The O-curve maximum (green square) under TDG perturbation remains close to the endogenous control (red circles). (**g**) Model (solid lines) provides good fits to the O-curves ($R^2 = 0.94$, $P = 10^{-19}$; see Supplementary Fig. 4a). Error bars are s.e. of the mean from 3 day–day repeats.

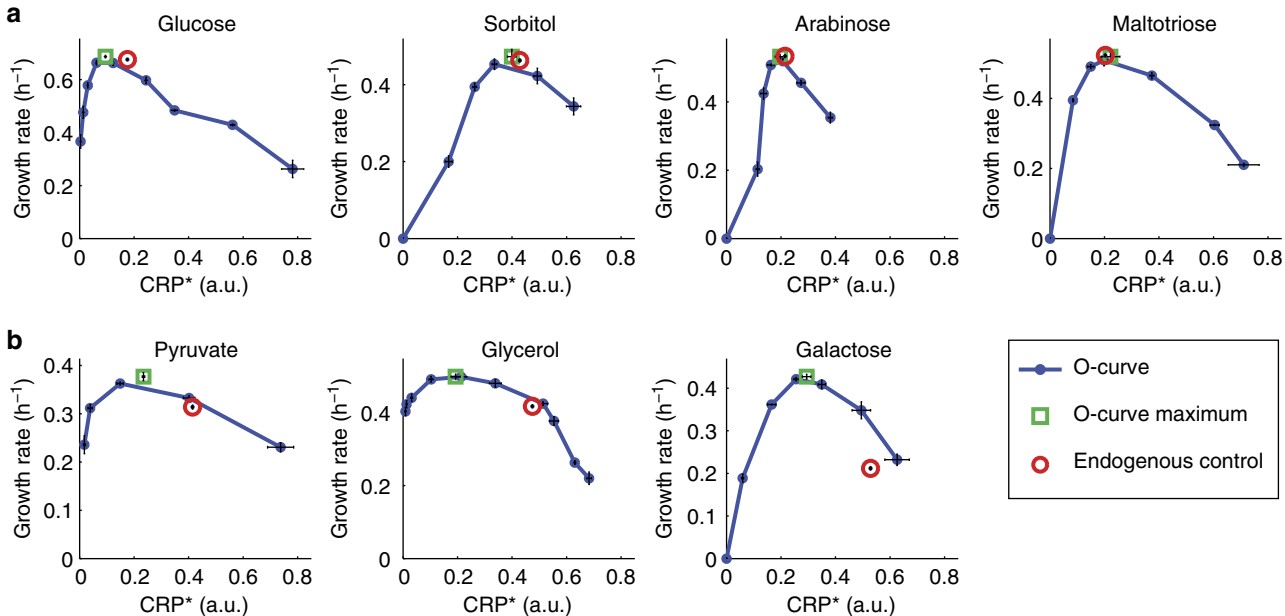

**Figure 2 | The carbon growth law does not maximize the growth rate under all conditions.** O-curves were measured on indicated carbon sources (blue). Red circles: endogenous control point. Green squares: O-curve maximum. (**a**) Endogenous cAMP control nearly maximized the growth rate on glucose, sorbitol, arabinose and maltotriose (difference to O-curve maximum <5%). (**b**) The growth rate was sub-optimal (difference to O-curve maximum >5%) on pyruvate, glycerol and galactose ($P<0.003$, one-sided $t$-test; Supplementary Table S1). For all conditions (optimal and sub-optimal) the endogenous control point was on the C-line (Fig. 1c; Supplementary Fig. 1c). Error bars are s.e. of the mean from 3 day-day repeats.

curvature at high C, fits the experimental C-line data well ($R^2 = 0.97$; Fig. 1c; Supplementary Fig. 4b).

The model allows us to ask when the growth law (C-line) maximizes the growth rate. We call CRP control robustly optimal if for all environmental conditions (represented by $\beta$ values) the C-line intersects the O-curve at its maximum (Fig. 3b), as is the case for lactose (Fig. 1).

It can be shown that control is robustly optimal if $P(C) = C$ and $k_1 = k_2 = k_f$ (Methods). In other words, robust optimality occurs when the expression of import enzymes is proportional to CRP activity, as occurs in the lactose system (Supplementary Fig. 2; ref. 33) and when the half-way points of ribosome saturation, allosteric regulation and cAMP inhibition by $x$ are matched. Best-fit parameters for lactose are close to this matching (Supplementary Fig. 5). We conclude that cells are able in principle to achieve optimal growth under all conditions in our model by measuring internal carbon concentration $x$ and using an appropriate control function $f(x)$.

**Optimal growth requires proportional control of carbon pumps.** The model also suggests ways to abolish optimality. One way is to change the control function $f(x)$ (Fig. 3c), which can be done by modulating levels of the phosphodiesterase $cpdA$ that degrades cAMP. Deletion of $cpdA$ indeed leads to sub-maximal growth rate (Supplementary Fig. 6).

Another way to abolish optimality is to make the relationship between a given sugar system expression and CRP* not strictly proportional, that is $P(C) \neq C$. Many sugar systems have a proportional relationship $P(C) = C$ (ref. 33), such that without CRP* ($C = 0$), no growth can be supported. However, some sugar systems, such as glycerol and pyruvate can support growth without cAMP (Supplementary Fig. 7a), so that $P(0) = C_0$. This occurs when the input function has a non-zero $y$ intercept, for example $P(C) = C + C_0$. In the model, such a non-zero $y$ intercept of $P(C)$ shifts the maximum of the O-curve away from the C-line

towards lower CRP* levels, leading to sub-optimal control (Fig. 3d; Supplementary Figure 7b), similar to that experimentally observed (Fig. 2b).

To test experimentally if a $y$ intercept of catabolic gene expression is sufficient to cause sub-optimal control we employed the naturally optimal sorbitol system. We expressed the sorbitol-specific transporter and catabolic enzymes ($srlAEBD$) under the inducible $Tet$ promoter, leaving the endogenous $srlAEBD$ genes intact. CRP* was nearly optimal when the $Tet$ promoter was repressed, similar to the wild-type strain (Fig. 4a, middle). However, when we induced the $Tet$ promoter by adding the inducer aTc the growth rate maximum increased and shifted to lower CRP* (Fig. 4a, right). This behaviour is quantitatively consistent with model predictions (Fig. 3d; Supplementary Fig. 8a; Supplementary Table 2), and turns the optimal sorbitol system into a sub-optimal design similar to glycerol and pyruvate (compare Fig. 4a and Fig. 2b).

**Sub-optimality for non-monotonically controlled carbon pumps.** According to the model, optimality also breaks down if catabolic genes are under control of a non-monotonic input function, such as $P(C) = C/(1 + (C/C_{max})^2)$ (Fig. 3e). Such a non-monotonic input function was observed for the galactose catabolism operon $galETK$ and for the galactose transporter $galP$[20]. Their expression peaks at intermediate cAMP concentrations due to regulation by an incoherent feed-forward loop (I-FFL). To test if the non-monotonic input function explains sub-optimality, we broke the I-FFL by deleting the $galS$ repressor, leading to a monotonic input function for $galETK$ and $galP$[36]. As predicted by the model, the $galS$ mutation made CRP* control optimal in galactose (Fig. 4b; Supplementary Fig. 8b; Supplementary Table 2), thus turning a non-optimal system into an optimal one.

**Sub-optimality of ppGpp synthesis.** We finally asked whether the control of another regulatory molecule, ppGpp, might also be

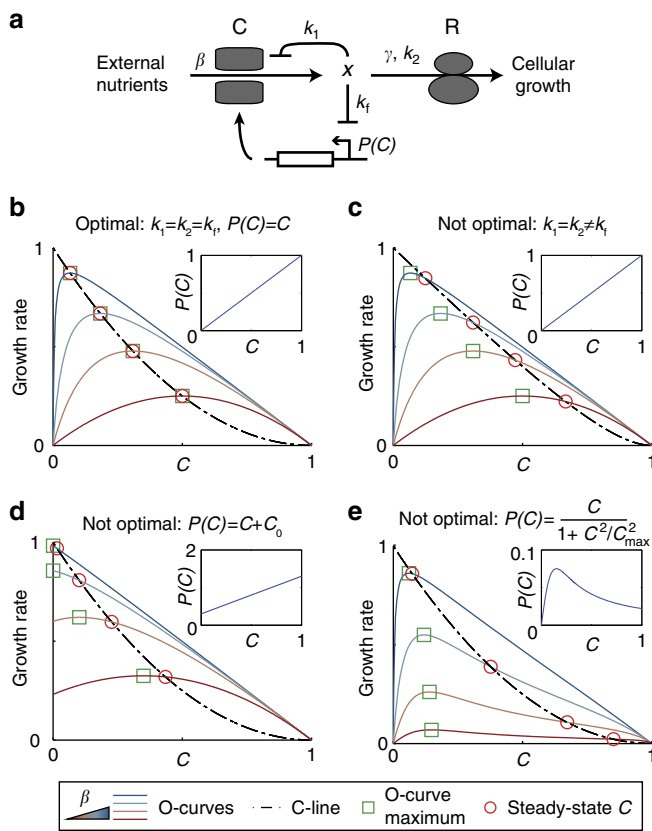

**b** Optimal: $k_1 = k_2 = k_f$, $P(C) = C$

**c** Not optimal: $k_1 = k_2 \neq k_f$

**d** Not optimal: $P(C) = C + C_0$

**e** Not optimal: $P(C) = \dfrac{C}{1 + C^2/C_{max}^2}$

Legend: $\beta$ — O-curves · — C-line □ O-curve maximum ○ Steady-state $C$

**Figure 3 | Model suggests conditions where growth is optimal or sub-optimal.** (**a**) A schematic representation of the model. Internal carbon $x$ is imported and catabolized by carbon sector $C$ at maximal rate $\beta$. $x$ inhibits carbon uptake with half-inhibition constant $k_1$. $\beta$ is high for good carbon sources and low for poor carbon sources. $x$ is consumed by the biomass production sector $R$. Given limited resources $R = 1 - C$. The maximal consumption rate of $x$ is $\gamma$ and $R$ is half-saturated with $x$ at $k_2$. Since $x$ inhibits cAMP which activates CRP which activates C-sector promoters, carbon catabolite repression is modelled by repression of C-sector genes by $x$ with half-inhibition constant $k_f$. The input function of the limiting enzyme for carbon uptake is given by $P(C)$ with $P(C) = C$ for most genes, due to proportional control of CRP targets. (**b**) Optimal control in the model is guaranteed when $k_1 = k_2 = k_f$ and $P(C) = C$ (Methods): the maxima of O-curves lie on the C-line (dotted black line). The maximum of the O-curve increases with $\beta$. (**c**) Control is sub-optimal when the transcriptional feedback strength is changed (modelled by $k_f$). This prediction agrees with experimental tests in Supplementary Fig. 6. (**d**) Control is sub-optimal for carbon uptake/catabolism genes that have non-proportional input functions ($P(C) \neq C$), such as genes with non-zero basal, CRP-independent expression ($P(C) = C + C_0$, inset). Such a $y$ intercept is observed for glycerol and pyruvate (Fig. 2b) that have sub-optimal control. The model predicts that the magnitude of sub-optimal control diminishes with increasing $\beta$, explaining why sub-optimal control is not apparent on glucose—which also has a non-zero intercept—within the experimental error (Fig. 2a). (**e**) Control is sub-optimal also for carbon uptake/catabolism genes with non-monotonic input functions (such as $P(C) = \frac{C}{1 + (\frac{C}{C_{max}})^2}$, inset). Sub-optimal control of CRP* on galactose is due to non-monotonic input functions of galETK and galP genes (Figs 2b and 4b). Model parameters: $\gamma = 1$, $k_1/k_2 = k_f/k_2 = 1$ (except for (panel c), where $k_f/k_2 = 4$). $\beta = 200, 20, 5, 1$; $C_0 = 0.3$; $C_{max} = 0.15$.

sub-optimal in some conditions and optimal in others. ppGpp is involved in the bacterial response to environmental change[37], including the linear scaling of the ribosomal proteomic fraction

with the growth rate under different growth conditions[8,27,38–40]. To test if endogenous ppGpp concentrations always maximize the growth rate we mutated one of the ppGpp synthetases, *relA*, and tested growth under the eight carbon sources studied above. We find that deletion of *relA* increases the growth rate in pyruvate, galactose and glycerol, indicating that ppGpp concentration is not always tuned to maximize growth rate. Unlike cAMP, ppGpp concentration was also sub-optimal on lactose (Supplementary Fig. 9).

## Discussion

To ask if and how cells compute optimal gene expression in diverse environments, we analysed how *E. coli* controls CRP activity on different carbon sources. Regulation of CRP activity is known to involve feedback inhibition by intracellular carbon[4,19,31]. We show here that this feedback mechanism optimally adjusts CRP activity for growth under many conditions, including several carbon sources and a wide range of perturbations to lactose uptake (Fig. 1; Supplementary Fig. 2). However, on several other carbon sources (that is, galactose, pyruvate and glycerol) the same feedback leads to sub-maximal growth rate (Fig. 2b).

Using a simple mathematical model, we show that robust optimal control of resource allocation requires proportionality between CRP activity and the expression of carbon catabolic enzymes (Fig. 3). We verified this prediction experimentally by genetic re-engineering of carbon gene control circuits. Introduction of a $y$ intercept in sorbitol gene expression, which abolishes proportionality, turned an optimal system into a sub-optimal one (Fig. 4a), and removing the non-monotonicity of galactose genes turned the sub-optimal control of CRP on galactose to be optimal (Fig. 4b).

Why did *E. coli* evolve non-proportional gene control functions that lead to sub-optimal allocation of bacterial resources (Fig. 2)? One possibility relates to evolutionary tradeoffs[41]: non-proportional control circuits can be beneficial under conditions other than the ones studied here. The I-FFL in the galactose system, for example, accelerates the activation of galactose genes[42], which may be beneficial when conditions change frequently. A $y$ intercept in pyruvate uptake genes allows for co-consumption of pyruvate together with carbon sources that cause low CRP activity, such as lactose, thereby increasing growth rate[43]. Indeed, we found that addition of pyruvate, but not of sorbitol, increases the growth rate on lactose and arabinose (Supplementary Fig. 10)[43]. More complex circuitry could presumably make the control of galactose and pyruvate genes optimal under a wider range of conditions. But such higher complexity would require additional regulators, which would incur a fitness cost[28,44].

The present approach may be used to search for other cases of sub-optimal control in biological regulatory circuits that need to respond to multiple inputs. Understanding such sub-optimality may reveal new aspects of the control mechanisms and suggest ways to usefully manipulate the behaviour of cells.

## Methods

**Strains and plasmids.** All experiments were done in MG1655 (CGSC #8003) background. Sub-optimality in pyruvate, glycerol and galactose was not due to a deletion around the *fnr* gene reported for this clone of MG1655 (ref. 45), as MG1655 (CGSC #6300), which is *fnr*[+], showed a similar degree of sub-optimality on these carbon sources (Supplementary Fig. 11). All deletion alleles were transduced using P1 phage from the Keio knockout collection[46], except the *lacI* mutation, which was made by homologous recombination[47]. The kanamycin resistance gene was removed using pCP20 (ref. 47). The *srlAEBD* operon was amplified from genomic DNA using primers oBT166 and oBT168 and cloned into HindIII/NcoI sites of pZA31 (ref. 48), which was amplified by PCR using primers

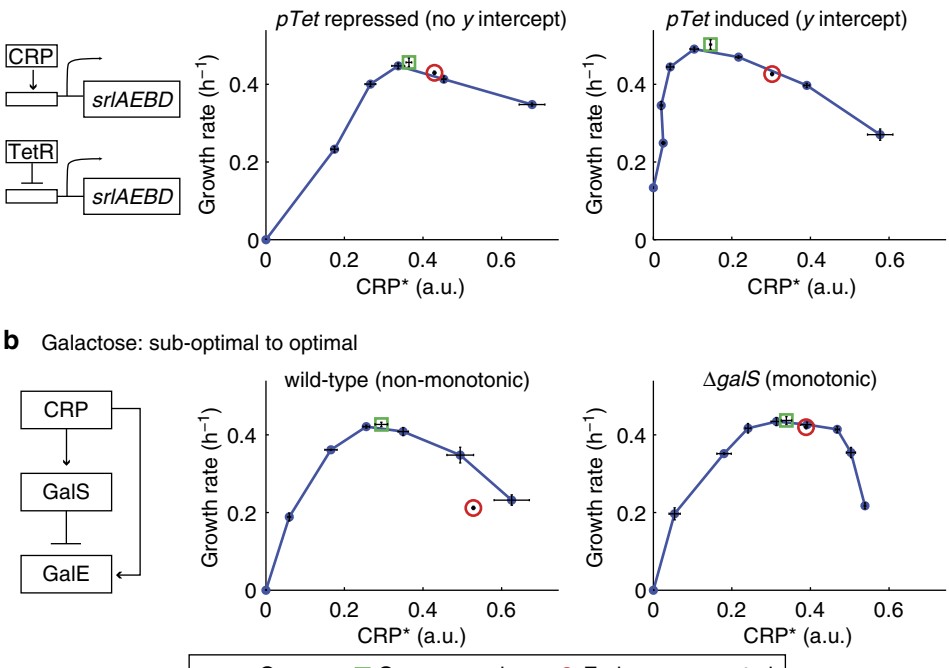

**Figure 4 | Re-engineering of carbon gene input functions fixes or disrupts optimal CRP control. (a)** Introduction of a CRP-independent copy of sorbitol genes (*srlAEBD* operon) under *pTet* control makes the input function have a non-zero basal level, when inducer (aTc) is added. Induction (125 ng ml$^{-1}$ aTc) results in sub-optimal control as predicted in Fig. 3d (Supplementary Fig. 8; Supplementary Table 2), whereas control is nearly optimal without induction. **(b)** Disruption of incoherent feed-forward loop (I-FFL) by deleting *galS* turns galactose input function (*galETK, galP* operons) from non-monotonic to monotonic, and turns control from sub-optimal to nearly optimal. This matches model prediction (Fig. 3e). Error bars are s.e. of the mean from 3 day-day repeats.

oBT202 and oBT203. Strains, plasmids and primers used in this study are listed in Supplementary Tables 3–5.

**Growth conditions.** All experiments were done in M9 minimal medium (42 mM Na$_2$HPO$_4$, 22 mM KH$_2$PO$_4$, 8.5 mM NaCl, 18.5 mM NH$_4$Cl, 2 mM MgSO$_4$, 0.1 mM CaCl$_2$, no uracil or thiamine) supplemented with appropriate antibiotics. The concentration of all carbon sources was 0.2% (w/v), except glycerol which was 0.2% (v/v) and pyruvate in Supplementary Fig. 10a, which was 0.1% (w/v). The nutrient concentrations ensure saturation of nitrogen and carbon[49], ranging between 4 mM (maltotriose) and 27 mM (glycerol). For O-curve and C-line measurements wild-type and *cyaA cpdA* mutant cells were inoculated in M9 + glucose from frozen glycerol stocks and pre-cultured by incubation overnight in the absence of cAMP (16–18 hours). This overnight culture was diluted 1:500 into 150 µl M9 medium containing one of the studied carbon sources and the indicated concentrations of other constituents (for example, cAMP) and the medium was overlaid with 100 µl Mineral oil (Sigma). GFP and optical density (OD) were recorded every 9 min in a robotic multi-well fluorimeter (Evoware, Tecan Infinite F200)[20]. Experiments were carried out in parallel on three different reporter strains: a CRP reporter, a GFP reporter of a constitutive σ70 promoter and a promoter-less GFP reporter used for background subtraction[20]. cAMP concentrations used for O-curves were 10, 5, 2.5, 1.25, 0.625, 0.31, 0.15, 0.078, 0.039 or 0 mM cAMP. Thio-di-glucoside (Santa Cruz sc-285346) concentrations were 1, 0.5 and 0.25 mM. For sorbitol *y* intercept experiments (Fig. 4a) the medium was supplemented with 0.25 mM isopropyl-β-D-thiogalactoside to alleviate the effects of high expression of *lacI* in this strain (due to transgenic *lacIq*), which unexpectedly affected the activity of the CRP reporter. The *Tet* promoter was induced by 125 ng ml$^{-1}$ anhydrotetracycline. The time interval between OD measurements for the comparison of wild-type and *relA* mutant (Supplementary Fig. 9) was only 4.5 min because GFP was not measured in this experiment. Cultures for the comparison of wild type and *relA* were pre-grown in LB, followed by overnight incubation in M9 + glucose. This overnight culture was used at a dilution of 1:500 for growth in 96-well plates in M9 supplemented with indicated carbon sources as described for O-curve and C-line measurements.

**Computation of growth rate and CRP activity.** For O-curve and C-line measurements, mid-exponential growth was automatically identified as the time point with the steadiest growth rate (Supplementary Fig. 1c), as follows: we first computed the growth rate by the time derivative of the logarithmic OD averaged over a 2-hour window (15 measurement points). To find the point of steadiest

growth we plotted the growth rate at 30 points equally spaced in log(OD) between OD = 0.001 and OD = 0.1. This transformation to OD-space made the algorithm more robust to experimental variations, such as small differences in the lag phase between technical repeats. To find the OD at mid-exponential growth (OD$_{mid-exp}$), we identified the minimum of the s.d. of the growth rate within a running window of 13 log(OD) spaced points (results were robust to changes of the window size; Supplementary Fig. 1d). To minimize experimental noise from points where gfp signal promoter activity was low, a minimum of OD = 0.01 was set. Promoter activity (PA = dGFP/dt/OD) of each reporter was averaged over a 2-hour window centred at the point of mid-exponential growth. CRP activity was defined as the PA of the CRP reporter divided by the PA of the σ70 reporter. The PA of the σ70 reporter scales linearly with the growth rate, which is expected for a constitutive promoter at the growth rates studied here[50] (Supplementary Fig. 12). Normalization by the σ70 reporter therefore ensures that our measurement of CRP activity represents the fraction of cellular resources dedicated to carbon catabolism.

**Measurement of co-consumption of lactose and pyruvate.** MG1655 (CGSC #8003) cells carrying the empty vector pU66 were grown o/n in M9 + lactose (0.2%) + kanamycin (50 µg ml$^{-1}$) and diluted 1:100 in 40 ml M9 + lactose (0.2%) + pyruvate (0.1%) + kanamycin (50 µg ml$^{-1}$). Cultures were grown with aeration at 37 °C, and samples were collected every hour followed by separation of bacteria from the medium using a 0.22 µm filter (Millipore). Pyruvate and lactose concentrations in the medium were measured using an Agilent 1,200 series high-performance liquid chromatography system (Agilent Technologies, USA) equipped with an anion exchange Bio-Rad HPX-87H column (Bio-Rad, USA). The column was eluted with 5 mM sulfuric acid at a flow rate of 0.6 ml min$^{-1}$ at 45 °C.[51]

**Criteria for robust optimal control of the C-sector.** This section shows that in the model, given the condition $k_1 = k_2 = k_f = k$ and $P(C) = C$, the control of C-sector size is optimal for all values of carbon uptake rate $\beta$.

The growth rate is proportional to the biomass production rate, such that given $R + C = 1$

$$\mu = \gamma(1 - C)\frac{x}{k + x}. \tag{1}$$

At steady state, the production rate and removal rate of $x$ are equal:

$$\beta C \frac{k}{x + k} = \gamma(1 - C)\frac{x}{k + x}. \tag{2}$$

Combining equations (1) and (2) gives the steady-state growth rate $\mu$ as a function of C (that is, the O-curve).

$$\mu(C) = \frac{(1-C)C\beta\gamma}{C(\beta-\gamma)+\gamma}, \tag{3}$$

$\mu(C)$ has a single maximum $\mu^* = \frac{\beta\gamma}{(\sqrt{\beta}+\sqrt{\gamma})^2}$ in the interval for $<0<C<1$ at $C^*$:

$$C^* = \frac{\sqrt{\beta\gamma}-\gamma}{\beta-\gamma}. \tag{4}$$

The regulatory feedback of $x$ on $C$ is given by:

$$C = f(x) = \frac{k}{k+x} \tag{5}$$

Combining equations (5) and (1) gives the steady-state solution for $C$.

$$C_{st} = \frac{\sqrt{\beta\gamma}-\gamma}{\beta-\gamma} \tag{6}$$

Comparing equations (4) and (6) shows that $C_{st} = C^*$ for all values of $\beta$, providing the optimal growth rate.

**Code availability**. Code used for analysis in Matlab (v2012a) and Wolfram Mathematica (v9) is available from the corresponding author (U.A.) on request.

**Data availability**. Data for C-line and O-curves are available in Supplementary Data sets 1–5. All other data is available from the corresponding author (U.A.) on request.

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

## Acknowledgements

We thank Ron Milo, Shalev Itzkovitz and members of the Alon group for careful reading of the manuscript and helpful discussions. We thank Shmuel Gleizer for help with measurements of metabolite concentrations, Avi Mayo for help in data analysis, and Tamar Danon for technical assistance. This work was supported by the Israel Science Foundation. U.A. is the incumbent of the Abisch-Frenkel Professorial Chair. B.D.T. thanks the Human Frontiers Science Project, the Swiss National Science Foundation, and the Society of the Swiss friends of the Weizmann Institute for a postdoctoral fellowship. Y.K. is supported by the Adams Fellowships Program of the Israel Academy of Sciences and Humanities.

## Author contributions

Study design: B.D.T., A.B., U.A. and R.S.; Execution of experiments: B.D.T.; Data analysis: B.D.T., Y.K. and S.D.; Writing manuscript: B.D.T., Y.K. and U.A.; Editing manuscript: B.D.T., Y.K., U.A., and A.B.; and Derivation of theory: B.D.T., Y.K., and U.A.

## Additional information

**Competing financial interests:** The authors declare no competing financial interests.

