## [Peer Review File · Nature Communications]

Reviewers' Comments:

Reviewer #1 (Remarks to the Author)

The manuscript of Towbin et al. investigates the optimality of gene expression regulation in bacteria by experimentally modulating it. Specifically, it asks whether the relationship between growth rate and the expression of genes involved in nutrient uptake and carbon catabolism reflects regulatory behaviours that maximize fitness. While the topic of growth-rate dependent proteome allocation has received considerable attention in recent years, no study, to my knowledge, has demonstrated the optimality of such growth laws through experimentally manipulating them. Furthermore, it remained unexplored whether growth laws are optimal under all conditions or whether they represent a simple heuristic that often works but sometimes fail. The manuscript therefore investigates a novel and important research question. The paper is well-written and clear. I found the analyses elegant and convincing and have only relatively minor comments as follows:

1) The manuscript is slightly confusing with respect to the link between growth law and rule-of-thumb solution. Apparently, multiple definitions of 'growth law' are offered. First, it is introduced as the linear relationship between growth rate and protein expression levels of nutrient uptake, catabolic and biomass producing enzymes:

"E. coli partitions its resources according to simple linear rules, called growth laws: The expression of proteins for biomass synthesis, such as ribosomes, increases linearly with growth rate. Conversely, the expression of enzymes for nutrient uptake and catabolism decreases approximately linearly with growth rate."

Based on this definition, the expression of proteins involved in pyruvate, glycerol and galactose uptake/catabolism is suboptimal, because it does not follow the same growth law as the expression of catabolic enzymes for other nutrients. Strictly speaking, they are not sub-optimal because of using the same rule-of-thumb solution, but rather because of their different ways of regulation, leading to non-proportional relationships with CRP and hence suboptimal growth rates.

However, later in the manuscript, 'growth law' is used more restrictively, referring to the approximately linear relationship between CRP activity level and growth rate (C-line). The presented results indeed show that the activity of this single gene is not optimal for all nutrients under the tested conditions. Therefore, the second definition of 'growth law' is preferred over the first one in order to formulate the tested hypothesis more clearly.

2) The Method section contains a laboratory evolution paragraph that is not even mentioned in the manuscript text. As I understand, the authors used a lab evolution to obtain a lactose transporter with a reduced import rate. The rationale of this somewhat convoluted approach should be better explained. Also, it appears from the Method that lacY mutant was created in a lacI deletant strain, which is in apparent contrast to a claim on page 5 "...over-expression of lactose metabolic genes by deletion of the lac repressor (LacI) or an impairment of LacY function by mutation were both matched by a shift of the O-curve along the C-line that yielded nearly optimal growth response."

3) Ref #29 appears to be incomplete.

Reviewer #2 (Remarks to the Author)

Summary of the key results

The authors test the hypothesis that cells use heuristic calculations, or "rules of thumb", that often work but sometimes fail, in contrast to accurate calculations allowing cells to be growth-optimal under all conditions. They find that *Escherichia coli* reaches optimal growth rate for 5/8 carbon sources. Growth optimality of wild-type *E. coli* was determined by experimentally varying CRP activity with exogenous cAMP in a $\Delta\text{cyaA}\Delta\text{cpdA}$ strain and comparing the optimal CRP level giving maximum growth rate with wild-type CRP activity and growth rate. They further find that having a non-zero y-intercept or non-monotonic protein concentration as a function of CRP-regulated proteome leads to suboptimal growth rate.

Originality and interest: if not novel, please give references

While the cAMP-CRP system has been extensively investigated this study appears original in that it performed direct experimental modulation of CRP activity under 8 different carbon sources. Further, it showed that wild-type cells modulate CRP activity to be optimal (in some cases and not in others) as explored by direct control of cAMP levels.

Data & methodology: validity of approach, quality of data, quality of presentation

Methods are sound, data are of high quality, and presented well (suggested revisions are listed later).

Appropriate use of statistics and treatment of uncertainties

Suggestions were made to improve appropriate use of statistics (please see list of revisions).

Conclusions: robustness, validity, reliability

The main finding is that CRP is tuned to maximize growth under some but not all carbon sources. From this result, the authors conclude that cells use rules of thumb: a heuristic that often works but sometimes fails. The conclusion, as currently stated, seems more a restatement of the observations than a more broadly applicable finding. This may be because the conclusion uses vague wording, such as "often works" and "sometimes fails."

What is the "rule of thumb" that was actually identified? The fact that the experiments were limited to one objective (growth rate) as a function of one variable (CRP activity) makes the generalizability of this conclusion unwarranted. Re-wording or more explicit statement of the actual scope of the conclusions will be necessary.

Can the rule-of-thumb conclusion be distinguished from the possibility that while cells may seem to suboptimal for one objective (growth rate), they may in fact be optimal for multiple objectives as shown by Shoval et al., *Science*, 2012, 336:1157-1160)? A discussion on this point would be helpful.

Suggested improvements: experiments, data for possible revision

1) To prove the validity of their theoretical model, the authors conducted experimental manipulations to test their predictions: They engineer some strains to turn optimal growth to suboptimal and vice versa (Figure 4). This is indeed very impressive. However I have three questions: 1. Can the theory predict the O-curve in the engineered system, just as they did in Figure 1G? If so please provide the predicted curve. 2. To turn an optimal into a suboptimal system is probably less a validation than the other direction. After all, disrupting the system probably may make the cell grow worse in many possible ways, and the proposed mechanism is consistent, but may not be the only explanation of what they observed, unless the theory can predict how much less growth in the tet-promoter induced system (Figure 4A, right). 3. This makes

the manipulation on the other direction(turn suboptimal system into optimal) especially exciting, however there is only one case study in the paper, whereas in the opposite direction there are two. Maybe the authors can add another one? For example, the authors showed that by adding a y-intercept to the catabolic gene expression in sorbitol, they turned an optimal system into suboptimal, how about design an experiment to rescue optimal growth from suboptimal in the cases of pyruvate, glycerol or galactose?

2) Fig 2: are the differences in growth rate between 'O-curve maximum' and 'endogenous control' points statistically significantly different, given the experimental uncertainty? Please provide a statistical test. More generally, please provide the (quantitative) criterion used to define 'suboptimal' CRP control for the experimental results.

3) Throughout the text the authors used "good fit" to describe the relationship between their experimental results and theoretical predictions. Although it is true that in most of the cases the theoretical curve and experimental data points match well visually (an exception is Figure 1G, red dots and the curve), I still think some vigorous statistical tests should be used to justify the "good fit" claim.

4) if space permits, it would be better to explain the error bars directly in the captions of Fig. 2 and Fig. 4 as well, and not just in Fig. 1.

5) There is actually a lot of information about the suboptimal growth of E. coli on glycerol and how that can be changed through laboratory evolution. This body of literature should be of interest to explain or discuss one of the three failure cases reported. A discussion on this point would be helpful.

Clarity and context: lucidity of abstract/summary, appropriateness of abstract, introduction and conclusions

Other than suggested revisions above, the article is clearly written.

Recommendation

The broader-reaching conclusion (that cells use rules-of-thumb) is not adequately supported. The findings all point towards a deeper investigation of CRP and linear growth laws, as opposed to a discovery of more general bacterial control laws. Therefore, the conclusion will need to be more adequately supported to justify publication for the broad readership of Nat Commun.

Reviewer #3 (Remarks to the Author)

The authors describe an interesting study that indicates that the cAMP-CRP control system of E coli can maximise the growth rate of this bacterium, provided that particular conditions are met. When those conditions are not met, the cAMP-CRP control mechanism is suboptimal. Therefore, the authors refer to this control mechanism as a rule of thumb that works often but not always. The achieve those results by comparing a CRP-titratable E coli strain with the wild-type. A comparison of the CRP levels and the growth rates of the two strains then indicates whether the wild type is optimal, which occurs when it achieves the same growth rate as the maximal growth rate attained by the titratable strain.

I think that this work is novel, at least in the way that it is carried out by titrating a major transcription factor in E coli. Earlier work that indicates that E coli can likely maximise its growth rate by optimal gene expression does exist, in addition to Dekel & Alon's work. Some of this earlier work is theoretical, and is referenced, but also experimental work exists and this is not referenced; and I think those should be added: those concerning E coli are: titration of citrate synthase by Walsh & Koshland, titration of H-ATPase by Peter Jensen & Hans Westerhoff, and titration of nearly

all glycolytic enzymes in *L. lactis* by Peter Jensen.

Recently a paper by the Hwa lab appeared on the nitrogen integration capacity of the CRP system in *E. coli* and I think that this should be mentioned in the introduction. This is important information for the reader as the Alon paper takes a purely carbon perspective.

We have the following major points about the result section:

(1) The experimental methods are insufficiently described.

a. No mention is made of carbon concentrations used in any of the experiments. This information is important to the results and conclusions presented. For the conclusions to hold, concentrations of external substrates should be saturating for all uptake systems.

b. M9 medium composition is not defined, again an important bit of information needed to evaluate the assertion that carbon-processing bottlenecks results in sub-optimal states. See the next point for why this is crucial information to include.

c. The cAMP-CRP circuit described by the authors is considered to sense carbon-nitrogen ratio's (see Huergo and Dixon, 2015; You et al., 2013) as opposed to carbon "levels" only. This means that the biosynthesis flux bottleneck (R-sector, figure 3A) could be caused by a nitrogen-limitation rather than an overcapacity in carbon uptake. Of course the outcome would be the same, with intracellular carbon accumulating, but the cause is different. For example, a carbon flux bottleneck could be brought about in a setting where the extracellular carbon substrate is present at low, non-saturating levels (i.e. uptake rates are far from maximal), but extracellular nitrogen is absent or present at such low levels that the nitrogen-uptake flux is unable to match the (low) carbon flux. In this scenario cells may be better served to induce high-affinity nitrogen uptake systems, rather than decreasing expression of carbon transporters.

i. Could they indicate how much carbon and nitrogen was included in their formulation of M9 (concentrations for every carbon source, see a. above). Also whether other supplements were added: thiamine? Uracil?

ii. Could the authors comment on whether their model and its predictions depends on the nature of the flux bottleneck occurring in the R-sector.

d. Pre-culturing conditions should be included. The physiological outcome of a growth experiment can be dramatically influenced by pre-culturing conditions.

i. How were cells pre-cultured (medium composition) and how long were they pre-cultured before the start of microtitre growth experiments. E.g. for the o-curve experiments, where cells also pre-cultured with cAMP? Where pre-cultures performed using the carbon source included during growth rate and CRP-activity measurements, or where cells always pre-cultured on M9 + glucose?

(2) The authors show that addition of pyruvate in the presence of lactose, leads to increased growth rates. This is somewhat surprising, given previous demonstrations by the group of Terry Hwa (You et al., 2013. Coordination of bacterial proteome with metabolism by cyclic AMP signalling, Nature) that addition of Pyruvate or any other α -ketoacid (2-oxoglutarate, oxaloacetate and pyruvate) leads to inhibition of adenylate cyclase, which in turn will lower CRP levels. While it could be that cells now have additional access to unregulated Pyruvate uptake, this should lead to a reduction in the expression of genes from the Lac-operon, reducing lactose uptake. The increased growth rate is therefore surprising. Could the authors address their finding with respect to the results presented by You et al. (2013)?

(3) In the materials and methods, the authors explain that experiments were repeated with an *fnr+* background, to exclude effects of *fnr* deletion in the experimental strains. However, the results are not shown. I would like to be able to evaluate these results. The authors can (and should) include these results in the Supporting materials.

(4) Throughout the paper σ_{70} expression is used to normalize CRP expression. Could the authors provide expression curves for σ_{70} ? How does σ_{70} expression change in the open- and closed-loop systems. To what extent are the O-curves and C-lines determined by changes in σ_{70} as opposed to changes in CRP? It can be the case e.g. that the shapes of the curves are determined by the denominator (σ_{70}), rather than the numerator (CRP) - which would lead to a different interpretation of the data. If changes in σ_{70} is dominant, then resource allocation bottlenecks would not match the model assumptions.

(5) I find the explanation of how growth rates and CRP activity were calculated very confusing. Could the authors try to improve this explanation? Possibly a figure in the Supplementary materials will help to indicate the various windows and regions used in their calculations.

(6) The authors state pyruvate is co-consumed together with carbon sources that cause low CRP activity.

a. Did the authors measure co-consumption?

b. What does CRP do in this scenario? Does it go up down, or no change?

(7) I find the explanation of the model in the result unnecessarily complicated. When it is carefully explained than also the derivations in the appendix can be easily done in the text which would greatly help the reader in understanding the main points of the model, which are: i) the conditions when the growth law corresponds to optimal behaviour, i.e. when $k_1=k_2=k_f$. This derivation can be done much simpler and should be added to the main text I think, it is central to the argumentation. ii) this derivation then also shows that the growth rate law is no longer optimal when changes are made to $f(x)$ or when a C has a basal level independent of cAMP. So if the modelling text is shortened and written much clearer all readers will be able to appreciate the elegance and insightfulness of the model.

Minor questions and comments:

(1) When the authors refer to carbon levels, do they mean concentrations? If so, concentrations would be the preferred term.

(2) Could the authors reference this statement: "The trade-off between reaching optimal gene expression and the cost of computing such optimal solutions is common to most organisms." This will not be obvious or a given to many readers.

(3) regarding the Abstract:

- the authors suggest that organisms overcome 'computational challenges' which is likely a bit awkward phrasing for some readers. (Also see line 3 of the introduction and usage of 'calculation mechanism').

-in the fourth sentence it is suggested that all bacterial gene expression follows linear dependencies with the growth rate. This is not true, not even under the conditions of balanced growth to which this statement refers, which is also not mentioned. This sentence may therefore confuse the reader. I would write something like: "When steady-state bacterial gene expression at exponential growth is compared at different growth rates, they turned out to scale linearly. This relation has become known as a 'growth law'."

(4) a comment about the use of the term "rule of thumb". I understand why the authors choose this term but I also think that it is a bit unfortunate. It suggests that E coli cannot do better than it does, whereas the origins of the limited optimal working of the cAMP-CRP may in particular tradeoffs in the system associated with dynamic environments or nitrogen-carbon integration or this control strategy is highly adaptive and can be qualitatively changed with only a few or single mutations during evolution. What I mean to say, I guess, is that I find the rule of thumb scientifically unsatisfactory and would appreciate additional discussion about the origins of the limited optimal behaviour of the circuit and a more substantiation of the using the term "rule of thumb" when one thinks about a microorganism that is changes its phenotype in order to enhance its fitness.

Reviewer #1 (Remarks to the Author):

The manuscript of Towbin et al. investigates the optimality of gene expression regulation in bacteria by experimentally modulating it. Specifically, it asks whether the relationship between growth rate and the expression of genes involved in nutrient uptake and carbon catabolism reflects regulatory behaviours that maximize fitness. While the topic of growth-rate dependent proteome allocation has received considerable attention in recent years, no study, to my knowledge, has demonstrated the optimality of such growth laws through experimentally manipulating them. Furthermore, it remained unexplored whether growth laws are optimal under all conditions or whether they represent a simple heuristic that often works but sometimes fail. The manuscript therefore investigates a novel and important research question. The paper is well-written and clear. I found the analyses elegant and convincing and have only relatively minor comments as follows:

We thank this reviewer for this endorsement of the novelty, originality and importance of this study.

1) The manuscript is slightly confusing with respect to the link between growth law and rule-of-thumb solution. Apparently, multiple definitions of 'growth law' are offered. First, it is introduced as the linear relationship between growth rate and protein expression levels of nutrient uptake, catabolic and biomass producing enzymes:

"E. coli partitions its resources according to simple linear rules, called growth laws: The expression of proteins for biomass synthesis, such as ribosomes, increases linearly with growth rate. Conversely, the expression of enzymes for nutrient uptake and catabolism decreases approximately linearly with growth rate."

Based on this definition, the expression of proteins involved in pyruvate, glycerol and galactose uptake/catabolism is suboptimal, because it does not follow the same growth law as the expression of catabolic enzymes for other nutrients. Strictly speaking, they are not sub-optimal because of using the same rule-of-thumb solution, but rather because of their different ways of regulation, leading to non-proportional relationships with CRP and hence suboptimal growth rates.

However, later in the manuscript, 'growth law' is used more restrictively, referring to the approximately linear relationship between CRP activity level and growth rate (C-line). The presented results indeed show that the activity of this single gene is not optimal for all nutrients under the tested conditions. Therefore, the second definition of 'growth law' is preferred over the first one in order to formulate the tested hypothesis more clearly.

We thank the reviewer for this comment which led us to clarify in the revised MS what we mean by growth law. We define growth laws as the linear dependence on growth rate of the bulk of enzymes needed for a global task of cell metabolism. We give two examples, linear increase of ribosomes with growth rate and linear decrease of carbon catabolism proteins with growth rate. We clarify more carefully which growth law we mean in each case.

2) The Method section contains a laboratory evolution paragraph that is not even mentioned in the manuscript text. As I understand, the authors used a lab evolution to obtain a lactose transporter with a reduced import rate. The rationale of this somewhat convoluted approach should be better explained. Also, it appears from the Method that lacY mutant was created in a lacI deletant strain, which is in apparent contrast to a claim on page 5 "...over-expression of lactose metabolic genes by deletion of the lac repressor (LacI) or an impairment of LacY function by mutation were both matched by a shift of the O-curve along the C-line that yielded nearly optimal growth response."

We agree with the reviewer that this experiment is somewhat peripheral to the major conclusion of the paper. We therefore removed it from the manuscript.

3) Ref #29 appears to be incomplete.

We corrected the incomplete reference.

Reviewer #2 (Remarks to the Author):

Summary of the key results

The authors test the hypothesis that cells use heuristic calculations, or "rules of thumb", that often work but sometimes fail, in contrast to accurate calculations allowing cells to be growth-optimal under all conditions. They find that Escherichia coli reaches optimal growth rate for 5/8 carbon sources. Growth optimality of wild-type E. coli was determined by experimentally varying CRP activity with exogenous cAMP in a Δ cyaA Δ cpdA strain and comparing the optimal CRP level giving maximum growth rate with

wild-type CRP activity and growth rate. They further find that having a non-zero y-intercept or non-monotonic protein concentration as a function of CRP-regulated proteome leads to suboptimal growth rate.

Originality and interest: if not novel, please give references

While the cAMP-CRP system has been extensively investigated this study appears original in that it performed direct experimental modulation of CRP activity under 8 different carbon sources. Further, it showed that wild-type cells modulate CRP activity to be optimal (in some cases and not in others) as explored by direct control of cAMP levels.

We thank the reviewer for this endorsement on the originality of the study.

Data & methodology: validity of approach, quality of data, quality of presentation

Methods are sound, data are of high quality, and presented well (suggested revisions are listed later).

Appropriate use of statistics and treatment of uncertainties

Suggestions were made to improve appropriate use of statistics (please see list of revisions).

Conclusions: robustness, validity, reliability

The main finding is that CRP is tuned to maximize growth under some but not all carbon sources. From this result, the authors conclude that cells use rules of thumb: a heuristic that often works but sometimes fails. The conclusion, as currently stated, seems more a restatement of the observations than a more broadly applicable finding. This may be because the conclusion uses vague wording, such as "often works" and "sometimes fails."

What is the "rule of thumb" that was actually identified? The fact that the experiments were limited to one objective (growth rate) as a function of one variable (CRP activity) makes the generalizability of this conclusion unwarranted. Re-wording or more explicit statement of the actual scope of the conclusions will be necessary.

We thank the reviewer for this comment which helped us to add new data addressing generalizability. We also removed ambiguity from the discussion section.

As the reviewer suggested, we add data on an additional variable to the study. We tested for this purpose a second growth law- the linear dependence of ribosomes on growth rate. This growth law is controlled by the ppGpp regulatory molecule.

We find that on several carbon sources the growth rate can also be improved by deleting one of the ppGpp synthetase, *relA*, indicating that like cAMP also ppGpp levels are not always tuned to maximize the growth rate. We find the biggest relative improvement of growth rate for the carbon sources that were also suboptimal for cAMP.

We believe that this data offers a basis for some generality of the conclusion that growth laws can be sub-optimal for growth rate in some conditions and optimal in others.

We now added this to the revised results section on pg. 10:

"Deletion of the ppGpp synthetase *relA* improves growth rate on several carbon sources

We finally asked whether other growth laws might also be suboptimal in some conditions and optimal in others. We therefore tested a second growth law, in which ribosomal proteomic fraction increases linearly with growth rate^{8,27}. Ribosomal expression is controlled by the regulatory molecule ppGpp³⁷⁻⁴⁰. We mutated one of the ppGpp synthetases, *relA*, and tested growth under the eight carbon sources studied above. We find that growth rate increases in pyruvate, galactose, and glycerol, indicating that the ribosomal growth law is suboptimal under the same carbon sources as the carbon growth law. Unlike the carbon growth law, the ribosomal growth law was also suboptimal on lactose (Figure S9)."

a**b**
Figure S9. Deletion of the ppGpp synthase *relA* increases the growth rate on lactose, galactose, glycerol, and pyruvate. a. *E. coli* encodes two synthetases for ppGpp (a regulatory molecule that inhibits ribosome production): RelA and SpoT. SpoT is bifunctional and also degrades ppGpp. We deleted *relA* to reduce ppGpp production. *relA* is best known for its function in the stringent response, an acute inhibition of ribosome synthesis upon amino acid starvation (nutritional down-shift), but *relA* mutants also have mildly reduced basal levels of ppGpp during steady state growth³. b. Relative increase in growth rate of $\Delta relA$ compared to wild-type. The growth rate was significantly increased on lactose, galactose, glycerol, and pyruvate ($p = 0.008$; 0.012 ; $4.4e-5$; 0.0078 , respectively; $p > 0.05$ for other sugars, one-sided t-test). The exponential growth rate was computed from a linear fit to $\log(OD)$ during exponential growth between $OD=0.003$ and $OD=0.03$. Error bars are standard error of the mean from 3 day-day repeats.

Can the rule-of-thumb conclusion be distinguished from the possibility that while cells may seem to suboptimal for one objective (growth rate), they may in fact be optimal for multiple objectives as shown by Shoal et al., Science, 2012, 336:1157-1160)? A discussion on this point would be helpful.

We now added this point to the paper. We discuss how evolutionary tradeoffs might cause sub-optimality for a given output - growth rate - in a single carbon source in the revised Discussion section on pg. 11, and cite the suggested reference. In this context we provide data showing that pyruvate sub-optimality (due to y-axis non-zero intercept) allows better growth in a different condition: pyruvate+lactose or arabinose (growth rate increase by 5% and 18%). We added new data showing that pyruvate is co-consumed with lactose. This data is shown in Figure S10.

Figure S10. Pyruvate is co-consumed with lactose and its addition improves growth rate on lactose and arabinose. a. Bacteria were grown in a medium containing a mixture of lactose (0.2%) and pyruvate (0.1%). The fraction of pyruvate and lactose remaining in the growth medium as a function of time, measured by HPLC, indicates co-consumption. Error bars are standard errors of the mean from 3 biologically independent cultures. b. Pyruvate, but not sorbitol, improves growth on lactose and arabinose. The growth rate of *E. coli* on lactose and arabinose (on which CRP activity is low) is improved by adding pyruvate to the growth medium (* $p < 0.05$, one-sided t-test). Shown is the relative difference of the growth rate on lactose (or arabinose) alone and when pyruvate (or sorbitol) is added. A positive difference indicates improved growth rate upon addition of the second carbon source. No significant improvement, or a decrease in growth rate, was observed for addition of sorbitol. c. Relative change of CRP

activity for same conditions as in b. d. The endogenous control point remains on the C-line in conditions with mixed carbon sources. The dotted line is the C-line model from Figure 1c. Error bars indicate standard error of the mean from 3 day-day repeats.

Suggested improvements: experiments, data for possible revision

1) To prove the validity of their theoretical model, the authors conducted experimental manipulations to test their predictions: They engineer some strains to turn optimal growth to suboptimal and vice versa (Figure 4). This is indeed very impressive.

We thank the reviewer for this endorsement.

However I have three questions:

1. Can the theory predict the O-curve in the engineered system, just as they did in Figure 1G? If so please provide the predicted curve.

We now show best-fit model curves to the engineered system data in Figure S8. We find the fit to be reasonably good, $R^2=0.74$ for the sorbitol engineered system and $R^2 = 0.93$ for the galactose engineered system. Statistical analysis of goodness of fit is shown in the legend of Figure S8.

a Sorbitol (y-intercept)

b Galactose (non-monotonic)

Figure S8. Model quantitatively predicts the O-curve of the engineered systems. O-curve data for engineered and native sorbitol (a) and galactose (b) systems from Figure 4 are shown with best fit model. The model for the native and engineered systems have equal values for all shared parameters (see Supplemental Text C). Error bars are standard errors of the mean from 3 day-day repeats. Quality of fit for sorbitol: $R^2 = 0.74$, p-val = $8.1e-5$, RMSE = 0.07/h, EV = 74%. Quality of fit for galactose: $R^2 = 0.93$, p-val = $1.4e-9$, RMSE = 0.05/h, EV = 92%.

2. To turn an optimal into a suboptimal system is probably less a validation than the other direction. After all, disrupting the system probably may make the cell grow worse in many possible ways, and the proposed mechanism is consistent, but may not be the only explanation of what they observed, unless the **theory can predict how much less growth** in the tet-promoter induced system (Figure 4A, right).

We thank the reviewer for this comment. We now show the model prediction to the engineered O-curve data for the *Tet*-promoter induced system in Figure S8. We find that the theory predicts a maximal growth rate of $\mu_{\max}=0.50$, which compares very well with the measured O-curve $\mu_{\max}=0.49\pm 0.01$. Thus, the reduction of the endogenous relative to the maximal growth rate is $14\pm 2\%$ in the model which compares very well with $13\pm 3\%$ in the measurement.

3. This makes the manipulation on the other direction (turn suboptimal system into optimal) especially exciting,

We thank the reviewer for this endorsement

However, there is only one case study in the paper, whereas in the opposite direction there are two. Maybe the authors can add another one? For example, the authors showed that by adding a y-intercept to the catabolic gene expression in sorbitol, they turned an optimal system into suboptimal, how about design an experiment to rescue optimal growth from suboptimal in the cases of pyruvate, glycerol or galactose?

We attempted to add a second example of turning a suboptimal system to be optimal but ran into technical difficulty.

To conduct a similar experiment to rescue optimal control of cAMP on glycerol, or pyruvate would indeed be a beautiful experiment. However, unfortunately, there is not enough molecular knowledge on the origin of the y-intercept in these systems to make such a manipulation feasible. The molecular nature of the pyruvate transporter remains unknown. Genetic data indicates that at least two transporters exist, one of which is expressed independently of CRP (Kreth et al., PlosONE 2013 and Ron Milo, personal communication). However, the molecular identity of these transporters is unknown and their identification would go beyond the scope of this manuscript.

Nevertheless, we believe that the two examples shown here (sorbitol y-intercept and galactose non-monotonicity) are strong support for our model predictions, especially with the added detailed comparison of the quantitative predictions of the theory.

2) Fig 2: are the differences in growth rate between 'O-curve maximum' and 'endogenous control' points statistically significantly different, given the experimental uncertainty? Please provide a statistical test. More generally, please provide the (quantitative) criterion used to define 'suboptimal' CRP control for the experimental results.

We thank the reviewer for this comment. We now provide a quantitative criterion in the text for suboptimality, which we define as a relative difference $>5\%$ between the endogenous control point and the O-curve maximum. We find that the points defined as suboptimal allow rejection of the null hypothesis that the growth rates are equal with p-value <0.03 , with most cases being <0.003 , using a one sided t-test. We now describe this statistics in Supplemental Table S1.

Table S1. Relative difference of growth rate at O-curve maximum and at the endogenous control point.

condition	$(\mu_{\max} - \mu_{\text{end}})/\mu_{\text{end}}$	Suboptimal (>5% difference)	p-value (one-sided t-test)
lactose	-3%	no	0.94
lactose + 0.25mM TDG	-1%	no	0.82
lactose + 0.5mM TDG	4%	no	0.049
lactose + 1mM TDG	3%	no	0.15
lactose (ΔlacI)	-1%	no	0.2
glucose	-2%	no	0.82
sorbitol	-2%	no	0.81
arabinose	-3%	no	0.77
maltotriose	-2%	no	0.71
pyruvate	15%	yes	0.0005
glycerol	20%	yes	0.0027
galactose	101%	yes	0.0004
galactose (ΔgalS)	4%	no	0.028
sorbitol + pZA31:srlAEBD (uninduced)	4%	no	0.12
sorbitol + pZA31:srlAEBD (125ng/ml aTC)	15%	yes	0.02

3) Throughout the text the authors used "good fit" to describe the relationship between their experimental results and theoretical predictions. Although it is true that in most of the cases the theoretical curve and experimental data points match well visually (an exception is Figure 1G, red dots and the curve), I still think some vigorous statistical tests should be used to justify the "good fit" claim.

We now add statistical tests of goodness of fit (Pearson correlation coefficient and associated p-value, root-mean-squared error, and explained variance) and report them in the legends of Figures S4, and S8. R^2 are between 0.73 and 0.94 with p-values are between $1e-19$ and $8e-5$.

Figure S4. Models of lactose O-curves and C-line fit measured data well. a. Correlation of measured growth rate on lactose (+TDG) and the growth rate predicted by the O-curve model (see Figure 1g). Correlation coefficient $R^2 = 0.94$, $p = 1.0e-19$, root-mean-squared error RMSE = 0.04/h, explained variance EV = 94%. b. Correlation of measured growth rate and the growth rate predicted by the C-line model (grey line in Figure 1c). Correlation coefficient $R^2 = 0.97$, $p = 6.9e-8$, RMSE = 0.03/h, EV = 92%. Error bars in x are the standard error of the mean (s.e.m.) of growth rate measurements (3 day-day repeats). Error bars in y are the s.e.m. of the growth rate predicted by the model from 3 day-day repeats.

4) if space permits, it would be better to explain the error bars directly in the captions of Fig. 2 and Fig. 4 as well, and not just in Fig. 1.

We added the definition of the error bars to all legends.

5) There is actually a lot of information about the suboptimal growth of *E. coli* on glycerol and how that can be changed through laboratory evolution. This body of literature should be of interest to explain or discuss one of the three failure cases reported. A discussion on this point would be helpful.

We now direct the reader to the glycerol evolutionary experiments in the paper. We believe that our results on y-intercept may add insight to previous findings of suboptimal growth on glycerol. On pg. 6 we now write:

"In the case of glycerol, our findings are consistent with evolutionary experiments that show that *E. coli* can rapidly evolve on glycerol to reach a faster growth rate with reduced cAMP concentrations^{13,15,30}"

Clarity and context: lucidity of abstract/summary, appropriateness of abstract, introduction and conclusions

Other than suggested revisions above, the article is clearly written.

Recommendation

The broader-reaching conclusion (that cells use rules-of-thumb) is not adequately supported. The findings all point towards a deeper investigation of CRP and linear growth laws, as opposed to a discovery of more general bacterial control laws. Therefore, the conclusion will need to be more adequately supported to justify publication for the broad readership of Nat Commun.

We now support the broader-reaching conclusion that growth laws can be sub-optimal by providing data that indicates that a second growth law (ribosome control by ppGpp) can be suboptimal for growth rate, with the satisfying result that it is suboptimal on almost exactly the same carbon sources as the carbon growth law. We also more clearly define our goals and conclusions throughout in order to enable a general readership to follow the paper accurately.

We hope that this new data and analysis and improved rigor makes the manuscript appropriate for Nat Commun.

Reviewer #3 (Remarks to the Author):

The authors describe an interesting study that indicates that the cAMP-CRP control system of *E. coli* can maximise the growth rate of this bacterium, provided that particular conditions are met. When those conditions are not met, the cAMP-CRP control mechanism is suboptimal. Therefore, the authors refer to this control mechanism as a rule of thumb that works often but not always. They achieve those results by comparing a CRP-titratable *E. coli* strain with the wild-type. A comparison of the CRP levels and the growth rates of the two strains then indicates whether the wild type is optimal, which occurs when it achieves the same growth rate as the maximal growth rate attained by the titratable strain.

I think that this work is novel, at least in the way that it is carried out by titrating a major transcription factor in *E. coli*.

We thank the reviewer for this endorsement.

Earlier work that indicates that *E. coli* can likely maximise its growth rate by optimal gene expression does exist, in addition to Dekel & Alon's work. Some of this earlier work is theoretical, and is referenced, but also experimental work exists and this is not referenced; and I think those should be added: those concerning *E. coli* are: titration of citrate synthase by Walsh & Koshland, titration of H-ATPase by Peter Jensen & Hans Westerhoff, and titration of nearly all glycolytic enzymes in *L. lactis* by Peter Jensen.

We have now added these references on earlier experimental work to the manuscript. We write on pg. 4: "To evaluate whether the C-line maximizes the growth rate we build on classic work employing titration of metabolic gene expression²¹⁻²³."

Recently a paper by the Hwa lab appeared on the nitrogen integration capacity of the CRP system in *E. coli* and I think that this should be mentioned in the introduction. This is important information for the reader as the Alon paper takes a purely carbon perspective.

We now mention in the introduction that the CRP system has also nitrogen integration capacity, and cited the study by the Hwa lab on pg. 4. We also refer to our recent study on growth in the presence of

poor nitrogen sources- which suggests that the carbon growth law itself can be inverted under certain poor nitrogen sources, such that glucose becomes one of the worst carbon sources for growth.

We have the following major points about the result section:

(1) The experimental methods are insufficiently described.

a. No mention is made of carbon concentrations used in any of the experiments. This information is important to the results and conclusions presented. For the conclusions to hold, concentrations of external substrates should be saturating for all uptake systems.

We now mention carbon concentrations throughout and note that carbon concentrations are saturating (0.2%, or between 4mM (maltotriose) - 27mM (glycerol)) .

b. M9 medium composition is not defined, again an important bit of information needed to evaluate the assertion that carbon-processing bottlenecks results in sub-optimal states. See the next point for why this is crucial information to include.

We now add the composition of the M9 medium to the methods section. We write "All experiments were done in M9 minimal medium (42 mM Na₂HPO₄, 22 mM KH₂PO₄, 8.5 mM NaCl, 18.5 mM NH₄Cl, 2 mM MgSO₄, 0.1 mM CaCl₂, no uracil or thiamine) supplemented with appropriate antibiotics."

c. The cAMP-CRP circuit described by the authors is considered to sense carbon-nitrogen ratio's (see Huergo and Dixon, 2015; You et al., 2013) as opposed to carbon "levels" only. This means that the biosynthesis flux bottleneck (R-sector, figure 3A) could be caused by a nitrogen-limitation rather than overcapacity in carbon uptake. Of course the outcome would be the same, with intracellular carbon accumulating, but the cause is different. For example, a carbon flux bottleneck could be brought about in a setting where the extracellular carbon substrate is present at low, non-saturating levels (i.e. uptake rates are far from maximal), but extracellular nitrogen is absent or present at such low levels that the nitrogen-uptake flux is unable to match the (low) carbon flux. In this scenario cells may be better served to induce high-affinity nitrogen uptake systems, rather than decreasing expression of carbon transporters.

i. Could they indicate how much carbon and nitrogen was included in their formulation of M9 (concentrations for every carbon source, see a. above). Also whether other supplements were added: thiamine? Uracil?

We now add the exact composition of the M9 medium to the methods section. The concentration of ammonia in this medium is 18.5 mM and at saturating levels. We found no measurable reduction in growth rate even when the ammonium concentration is reduced 10-fold (see also Bren et al., BMC Systems Biology 2013). No supplements (uracil, thiamine) were added.

ii. Could the authors comment on whether their model and its predictions depend on the nature of the flux bottleneck occurring in the R-sector.

This study considered the situation where nitrogen and other factors are saturating, and the flux bottleneck occurring in the R-sector is due to carbon. Future work can attempt to generalize the model to different flux bottleneck scenarios. It will be intriguing to see whether a control configuration can in principle be found that allows optimal growth rate over all variations of multiple limiting factors, similar to the $k_1=k_2=k_i$ condition found here for carbon alone.

d. Pre-culturing conditions should be included. The physiological outcome of a growth experiment can be dramatically influenced by pre-culturing conditions.

i. How were cells pre-cultured (medium composition) and how long were they pre-cultured before the start of microtitre growth experiments. E.g. for the o-curve experiments, were cells also pre-cultured with cAMP? Where pre-cultures performed using the carbon source included during growth rate and CRP-activity measurements, or were cells always pre-cultured on M9 + glucose?

We now add this information to the Methods section: cells were pre-cultured on M9 + glucose.

(2) The authors show that addition of pyruvate in the presence of lactose, leads to increased growth rates. This is somewhat surprising, given previous demonstrations by the group of Terry Hwa (You et al., 2013. Coordination of bacterial proteome with metabolism by cyclic AMP signalling, Nature) that addition of Pyruvate or any other α -ketoacid (2-oxoglutarate, oxaloacetate and pyruvate) leads to inhibition of adenylate cyclase, which in turn will lower CRP levels. While it could be that cells now

have additional access to unregulated Pyruvate uptake, this should lead to a reduction in the expression of genes from the Lac-operon, reducing lactose uptake. The increased growth rate is therefore surprising. Could the authors address their finding with respect to the results presented by You et al. (2013)?

The group of Terry Hwa found that addition of pyruvate lowers CRP activity (You et al., 2013)- there is a strong transient lowering, after which CRP activity recover to a slightly lower level than before (see relevant figure inserted below). In that paper growth rate is not reported in these conditions.

In a second paper (Hermsen et al., 2015), the Hwa group finds that adding pyruvate to several other carbon sources increases growth rate.

Our findings fully agree with the findings of the Hwa group. Our measurement of CRP activity is done in steady state growth and so agrees with the mild long-term reduction found by You et al., 2013, although a precise quantification of CRP activity is not provided by You et al. Our finding of increased growth upon pyruvate addition agrees with Hermsen et al., 2015.

We added a new Supplemental figure (Figure S10) addressing this point in more detail. In this section we show the data of CRP activity upon pyruvate addition.

You et al., Figure S26:

Figure S10. Pyruvate is co-consumed with lactose and its addition improves growth rate on lactose and arabinose. a. Bacteria were grown in a medium containing a mixture of lactose (0.2%) and pyruvate (0.1%). The fraction of pyruvate and lactose remaining in the growth medium as a function of time, measured by HPLC, indicates co-consumption. Error bars are standard errors of the mean from 3 biologically independent cultures. b. Pyruvate, but not sorbitol, improves growth on lactose and arabinose. The growth rate of *E. coli* on lactose and arabinose (on which CRP activity is low) is improved by adding pyruvate to the growth medium (* $p < 0.05$, one-sided t-test). Shown is the relative difference of the growth rate on lactose (or arabinose) alone and when pyruvate (or sorbitol) is added. A positive difference indicates improved growth rate upon addition of the second carbon source. No significant improvement, or a decrease in growth rate, was observed for addition of sorbitol. c. Relative change of CRP activity for same conditions as in b. d. The endogenous control point remains on the C-line in conditions with mixed carbon sources. The dotted line is the C-line model from Figure 1c. Error bars indicate standard error of the mean from 3 day-day repeats.

(3) In the materials and methods, the authors explain that experiments were repeated with an *fnr+* background, to exclude effects of *fnr* deletion in the experimental strains. However, the results are not

shown. I would like to be able to evaluate these results. The authors can (and should) include these results in the Supporting materials.

We now add this data to the supplemental material, in a new supplemental Figure S11.

Figure S11. Suboptimality on galactose, glycerol and pyruvate is not due to an *fnr* deletion.

Shown is the relative growth rate difference of the endogenous control point and the maximum of the O-curve for two different MG1665 isolates, with and without a deletion of the genomic region around *fnr*. Both strains show a similar degree of sub-optimality for growth on the indicated carbon sources. Error bars are standard errors of the mean from 3 day-day repeats.

(4) Throughout the paper σ_{70} expression is used to normalize CRP expression. Could the authors provide expression curves for σ_{70} ? How does σ_{70} expression change in the open- and closed-loop systems. To what extent are the O-curves and C-lines determined by changes in σ_{70} as opposed to changes in CRP? It can be the case e.g. that the shapes of the curves are determined by the denominator (σ_{70}), rather than the numerator (CRP) - which would lead to a different interpretation of the data. If changes in σ_{70} is dominant, then resource allocation bottlenecks would not match the model assumptions.

We thank the reviewer for this comment. We now provide full expression curves for σ_{70} in the open- and closed-loop systems, in a new SI section (Figure S12). We find that the promoter activity of the σ_{70} reporter scales accurately with growth rate. Such scaling is expected for the protein production rate from a constitutive promoter in the range of growth rates studied here (see also Gerosa et al., MSB 2013). Since our model deals with CRP-controlled expression as a fraction of total expression, it is essential to normalize by the promoter activity of the σ_{70} promoter. We now clarify this in the revised Methods section.

Figure S12. The activity of the σ_{70} reporter scales with the growth rate. Promoter activity of the σ_{70} reporter (black) and growth rate (red) normalized to the maximum in each condition are shown as a function of CRP activity for O-curves on indicated carbon sources and for the C-line (same carbon sources as in Figure S2a). Error bars are standard error of the mean from 3 day-day replicates.

(5) I find the explanation of how growth rates and CRP activity were calculated very confusing. Could the authors try to improve this explanation? Possibly a figure in the Supplementary materials will help to indicate the various windows and regions used in their calculations.

We now clarify the explanation of the algorithm used to determine the growth plateau and computation of the promoter activity and growth rate and add a figure (Figure S1). In the revised Methods we write:

“For O-curve and C-line measurements, mid-exponential growth was automatically identified as the time point with the most steady growth rate (see also Figure S1c), as follows: We first computed the growth rate by the time derivative of the logarithmic optical density (OD) averaging over a 2 hour window (15 measurement points). To find the point of most steady growth we plotted the growth rate at 30 points equally spaced in $\log(\text{OD})$ between $\text{OD}=0.001$ and $\text{OD}=0.1$. This transformation to OD-space made the algorithm more robust to experimental variations, such as small differences in the lag phase between technical repeats. To find the OD at mid-exponential growth ($\text{OD}_{\text{mid-exp}}$) we identified the minimum of the standard deviation of the growth rate within a running window of 13 $\log(\text{OD})$ spaced points (results were robust to changes of the window size, Figure S1d). To minimize experimental noise from points where promoter activity was low, a minimum of $\text{OD} = 0.01$ was set. Promoter activity ($\text{PA} = \text{dGFP}/\text{dt}/\text{OD}$) of each reporter was averaged over a 2 hour window centered around the point of mid-exponential growth. CRP activity was defined as the PA of the CRP reporter divided by the PA of the σ_{70} reporter. The PA of the σ_{70} reporter scales linearly with the growth rate, which is expected for the protein production rate from a constitutive promoter for the growth rates studied here⁵⁰ (Figure S12). Normalization by the σ_{70} reporter therefore ensures that our measurement of CRP activity represents the fraction of cellular resources dedicated to carbon catabolism.”

Figure S1. Measurement of CRP activity using a GFP reporter. a. CRP-cAMP activates several hundred genes, many involved in carbon catabolism and uptake. To measure CRP activity

we introduce a synthetic CRP-dependent GFP reporter on a low-copy plasmid¹. b. We measure CRP activity as the ratio of the promoter activity of a CRP dependent GFP reporter and an otherwise identical, but CRP independent GFP reporter. This approach normalizes for global changes in gene expression associated with changes in growth rate and for condition dependent plasmid copy-number. The activities of the reporters were measured independently in two strains (whose only difference is the reporter plasmid they carry). Auto-fluorescence background was subtracted using a strain with a promoterless GFP. The three strains were grown in parallel under identical conditions. GFP and optical density (OD) were measured every 9 minutes in a robotic plate reader and promoter activity was calculated as the rate of GFP production per optical density ($PA = dGFP/dt/OD$). c. approach used to determine mid-exponential growth. Growth rate was computed averaging over a 2 hour window (= 15 measurement points). To identify mid-exponential growth we plotted the growth rate as a function of $\log(OD)$ and identified the point with minimal standard-deviation of the growth rate within a running window of 13 points. The growth rate at this point was used for further analysis and the PA of CRP and $\sigma 70$ reporters was calculated from a 2 hour window centered around the same time point. d. The growth rate at the endogenous control point and at the O-curve maximum were computed using the approach outlined in (c) using window sizes between 7 and 19 points for computing the standard deviation (step 4). The measured growth rates are robust to variation of the window size. Error bars are standard errors of the mean from 3 day-day repeats.

(6) The authors state pyruvate is co-consumed together with carbon sources that cause low CRP activity.

a. Did the authors measure co-consumption?

We have added to the revised manuscript new data on co-consumption. We measured pyruvate and lactose in the medium using HPLC at different time points (see Figure S10) finding co-consumption.

b. What does CRP do in this scenario? Does it go up down, or no change?

We now show in Figure S10 that the CRP activity is slightly reduced upon addition pyruvate to lactose. This reduction of CRP activity is in agreement with the increase in growth rate under these conditions and a shift along the C-line (see also Hermsen et al., 2015).

(7) I find the explanation of the model in the result unnecessarily complicated. When it is carefully explained than also the derivations in the appendix can be easily done in the text which would greatly help the reader in understanding the main points of the model, which are: i) the conditions when the growth law corresponds to optimal behaviour, i.e. when $k_1=k_2=k_f$. This derivation can be done much simpler and should be added to the main text I think, it is central to the argumentation. ii) this derivation then also shows that the growth rate law is no longer optimal when changes are made to $f(x)$ or when a C has a basal level independent of cAMP. So if the modelling text is shortened and written much clearer all readers will be able to appreciate the elegance and insightfulness of the model.

We thank the reviewer for this positive evaluation of our modelling approach. We have shortened and clarified the description of the model in the results section and the SI, and added a methods section with the derivation of the conditions for optimality. This new section reads:

Derivation of criteria for robust optimal control of the C-sector

This section shows that in the model, given the condition $k_1 = k_2 = k_f = k$ and $P(C) = C$, the control of C-sector size is optimal for all values of carbon uptake rate β .

The growth rate is proportional to the biomass production rate, such that given $R + C = 1$

$$(1) \quad \mu = \gamma(1 - C) \frac{x}{k+x}$$

At steady state the production rate and removal rate of x are equal:

$$(2) \quad \beta C \frac{k}{x+k} = \gamma(1 - C) \frac{x}{k+x}$$

Combining eq. (1) and eq. (2) gives the steady-state growth rate μ as a function of C (i.e. the O-curve).

$$(3) \quad \mu(C) = \frac{(1-C)C\beta\gamma}{C(\beta-\gamma)+\gamma}$$

$\mu(C)$ has a single maximum $\mu^* = \frac{\beta\gamma}{(\sqrt{\beta}+\sqrt{\gamma})^2}$ in the interval for $0 < C < 1$ at C^* :

$$(4) \quad C^* = \frac{\sqrt{\beta}\sqrt{\gamma}-\gamma}{\beta-\gamma}$$

The regulatory feedback of x on C is given by:

$$(5) \quad C = f(x) = \frac{k}{k+x}$$

Combining eq. (5) and eq. (1) gives the steady-state solution for C.

$$(6) \quad C_{st} = \frac{\sqrt{\beta}\sqrt{\gamma}-\gamma}{\beta-\gamma}$$

Comparing eq. 4 and eq. 6 shows that $C_{st} = C^*$ for all values of β , providing the optimal growth rate.

Minor questions and comments:

(1) When the authors refer to carbon levels, do they mean concentrations? If so, concentrations would be the preferred term.

We replace carbon level with concentration throughout the text.

(2) Could the authors reference this statement: "The trade-off between reaching optimal gene expression and the cost of computing such optimal solutions is common to most organisms." This will not be obvious or a given to many readers.

We made the discussion on the rule-of-thumb more rigorous and removed this sentence.

(3) regarding the Abstract:

- the authors suggest that organisms overcome 'computational challenges' which is likely a bit awkward phrasing for some readers. (Also see line 3 of the introduction and usage of 'calculation mechanism').

We replace the term computational with regulatory and computing with finding.

- in the fourth sentence it is suggested that all bacterial gene expression follows linear dependencies with the growth rate. This is not true, not even under the conditions of balanced growth to which this statement refers, which is also not mentioned. This sentence may therefore confuse the reader. I would write something like: "When steady-state bacterial gene expression at exponential growth is compared at different growth rates, they turned out to scale linearly. This relation has become known as a 'growth law'."

Given 150-word abstract length constraint we adapted the suggested phrasing a little and write now: "The growth-rate bacteria reach on different carbon sources declines linearly with the steady-state expression of carbon-catabolic genes, a relation known as a bacterial growth-law."

We similarly revised the corresponding paragraph in the introduction.

(4) a comment about the use of the term "rule of thumb". I understand why the authors choose this term but I also think that it is a bit unfortunate. It suggests that E coli cannot do better than it does, whereas the origins of the limited optimal working of the cAMP-CRP may in particular tradeoffs in the system associated with dynamic environments or nitrogen-carbon integration or this control strategy is highly adaptive and can be qualitatively changed with only a few or single mutations during evolution. What I mean to say, I guess, is that I find the rule of thumb scientifically unsatisfactory and would appreciate additional discussion about the origins of the limited optimal behaviour of the circuit and a more substantiation of the using the term "rule of thumb" when one thinks about a microorganism that is changes its phenotype in order to enhance its fitness

In the revised manuscript we now minimize the use of the concept rule of thumb. We discuss the possibility that the limited optimal working of the CRP control system may result from a tradeoff between optimality and the cost of elaborate control circuits. For example, a non-zero y-intercept for pyruvate allows it to be co-consumed with other sugars enhancing growth, but leads to suboptimal growth rate when pyruvate is consumed alone. Optimality on pyruvate alone can be in principle achieved by an additional regulatory circuit that senses whether other carbon sources (e.g. lactose, arabinose) are present and accordingly modify the y-intercept- but such a mechanism is probably too costly to be selected given the presumed rarity of pyruvate as a sole carbon source.

In summary, we thank the reviewers whose comments helped us to add new data and analysis and to clarify the text. We believe that the revised paper is improved in terms of generality, rigor and clarity.

Reviewers' Comments:

Reviewer #1 (Remarks to the Author)

I found the revised version of the manuscript improved and ready for publication. The text is clearer now and the new section on the ribosomal growth law is an important addition to the story.

Reviewer #2 (Remarks to the Author)

The manuscript has been greatly improved based on the additional data, reporting of statistical tests, and clarification of key points.

I have but two remaining concerns, described below.

Otherwise, I am satisfied that my concerns have been addressed adequately.

1) One of my concerns was that the rules of thumb concept was not generalizable. In response, the authors added new data involving *relA* deletion to reduce ppGpp levels. The authors show that *relA* deletion increase growth rate on four carbon sources, where 3/4 carbon sources overlap with those improving growth for cAMP-CRP control.

I certainly appreciate this new data and the result itself is interesting.

However, there is a logical gap when trying to link the decreased growth rate to the additional growth law (linear dependence of ribosomes on growth rate).

The ppGpp alarmone has numerous effects besides ribosome production that may affect growth rate (Daelbroux & Swanson, doi:10.1038/nrmicro2720). Therefore, it seems difficult to justify that this experiment directly tests the second growth law referred to by the authors (linear dependence of ribosomes on growth rate).

For this reason, I suggest re-phrasing, justifying or excluding the statement concerning the second growth law.

I suggest keeping the statement that "ppGpp levels are not always tuned to maximize growth rate," as it is still valid.

2) Quoting from the previous response:

Question:

2. To turn an optimal into a suboptimal system is probably less a validation than the other direction.

After all, disrupting the system probably may make the cell grow worse in many possible ways, and the

proposed mechanism is consistent, but may not be the only explanation of what they observed, unless

the theory can predict how much less growth in the tet-promoter induced system (Figure 4A, right).

Author Response:

We thank the reviewer for this comment. We now show the model prediction to the engineered O-curve data for the Tet-promoter induced system in Figure S8. We find that the theory predicts a maximal growth rate of $\mu_{max}=0.50$, which compares very well with the measured O-curve $\mu_{max}=0.49\pm$

0.01. Thus, the reduction of the endogenous relative to the maximal growth rate is $14\pm 2\%$ in

the model which compares very well with $13 \pm 3\%$ in the measurement.

My remaining concern:

The authors' statement, "Thus, the reduction of the endogenous relative to the maximal growth rate is $14 \pm 2\%$ in the model which compares very well with $13 \pm 3\%$ in the measurement" if true, will address my concerns very well.

However I was not able to see how this conclusion was arrived at, and I also failed to see why Figure S8 directly addresses this question. What would be the key result in Figure S8 that addresses this question? I suggest clarifying this point further.

Reviewer #4 (Remarks to the Author)

I was asked to review this manuscript, with special attention to evaluating whether the responses to questions and issues raised by former Reviewer #3 were appropriately addressed in the revised manuscript.

In addition to corroborating the overall impression that this constitutes an interesting and original paper, I found that the authors adequately addressed all the issues raised by Reviewer #3.

One very minor point is that the authors mention the explicit carbon concentrations in the rebuttal letter (4mM (maltotriose) - 27mM (glycerol)), but I could not find explicit mention of these values in the manuscript. I would suggest to add the values, as requested by the reviewer.

Reviewers' comments:

Reviewer #1 (Remarks to the Author):

I found the revised version of the manuscript improved and ready for publication. The text is clearer now and the new section on the ribosomal growth law is an important addition to the story.

We thank this reviewer for this endorsement.

Reviewer #2 (Remarks to the Author):

The manuscript has been greatly improved based on the additional data, reporting of statistical tests, and clarification of key points.

We thank this reviewer for this endorsement.

I have but two remaining concerns, described below. Otherwise, I am satisfied that my concerns have been addressed adequately.

1) One of my concerns was that the rules of thumb concept was not generalizable. In response, the authors added new data involving *relA* deletion to reduce ppGpp levels. The authors show that *relA* deletion increase growth rate on four carbon sources, where 3/4 carbon sources overlap with those improving growth for cAMP-CRP control.

I certainly appreciate this new data and the result itself is interesting.

We thank this reviewer for this endorsement.

However, there is a logical gap when trying to link the decreased growth rate to the additional growth law (linear dependence of ribosomes on growth rate).

The ppGpp alarmone has numerous effects besides ribosome production that may affect growth rate (Daelbroux & Swanson, doi:10.1038/nrmicro2720). Therefore, it seems difficult to justify that this experiment directly tests the second growth law referred to by the authors (linear dependence of ribosomes on growth rate).

For this reason, I suggest re-phrasing, justifying or excluding the statement concerning the second growth law. I suggest keeping the statement that "ppGpp levels are not always tuned to maximize growth rate," as it is still valid.

This comment helped us to clarify the text. We add the suggested reference and write in the results section on pg. 10:

"We finally asked whether the control of another regulatory molecule, ppGpp, might also be suboptimal in some conditions and optimal in others. ppGpp is involved in the bacterial response to environmental change³⁷, including the linear scaling of the ribosomal proteomic fraction with the growth rate under different growth conditions^{8,27,40-42}. To test if endogenous ppGpp concentrations always maximize the growth rate we mutated one of the ppGpp synthetases, *relA*, and tested growth under the eight carbon sources studied above. We find that the growth rate increases in pyruvate, galactose, and glycerol, indicating that ppGpp concentration is not always tuned to maximize growth rate. Unlike cAMP, ppGpp concentration was also suboptimal on lactose (Figure S9)."

2) Quoting from the previous response:

Question 2. To turn an optimal into a suboptimal system is probably less a validation than the other direction. After all, disrupting the system probably may make the cell grow worse in many possible ways, and the

proposed mechanism is consistent, but may not be the only explanation of what they observed, unless the theory can predict how much less growth in the tet-promoter induced system (Figure 4A, right).

Author Response:

We thank the reviewer for this comment. We now show the model prediction to the engineered O-curve data for the Tet-promoter induced system in Figure S8. We find that the theory predicts a maximal growth rate of $\mu_{\max}=0.50$, which compares very well with the measured O-curve $\mu_{\max}=0.49\pm 0.01$. Thus, the reduction of the endogenous relative to the maximal growth rate is $14\pm 2\%$ in the model which compares very well with $13\pm 3\%$ in the measurement.

My remaining concern:

The authors' statement, "Thus, the reduction of the endogenous relative to the maximal growth rate is $14\pm 2\%$ in the model which compares very well with $13\pm 3\%$ in the measurement" if true, will address my concerns very well. However, I was not able to see how this conclusion was arrived at, and I also failed to see why Figure S8 directly addresses this question. What would be the key result in Figure S8 that addresses this question? I suggest clarifying this point further.

We are thankful for this comment and now clarify this point in the SI. We add a Supplementary Table 2 with the relevant data and highlight the relevant data points in Figure S8. The relevant growth rates and deviations from maximum are noted for both experiment and model.

We note a slight technical improvement in our data analysis that caused a small change on the order of 0.02 in the best fit growth rates (eg from 0.5/h to 0.48/h), which does not affect our conclusions. Whereas before we first averaged the data over 3 day-day repeats and then fitted the model to the averaged data, now, in order to get an error bar for the predicted O-curve maximum, we fitted the model to each day-day repeat individually and then took the mean and s.e.m. of the three day-day repeats. We mention this analysis now in Supplemental Table 2.

Supplementary Table 2. Comparison of measured and predicted growth rates for the engineered sorbitol and galactose systems

	Measured maximal growth rate on O-curve (μ_{\max}^{me}) [h^{-1}]	Predicted maximal growth rate on O-curve [§] (μ_{pr}^{\max}) [h^{-1}]	Growth rate at endogenous control point (μ_{end}) [h^{-1}]	Measured deviation from maximum ($(\mu_{\max}^{\text{me}} - \mu_{\text{end}}) / \mu_{\text{end}}$) [%]	Predicted deviation from maximum ($(\mu_{\text{pr}}^{\max} - \mu_{\text{end}}) / \mu_{\text{end}}$) [%]
Sorbitol pTet uninduced	0.45 +/- 0.01	0.42 +/- 0.01	0.43 +/- 0.004	4 +/- 2	3 +/- 2
Sorbitol pTet induced	0.49 +/- 0.01	0.48 +/- 0.01	0.43 +/- 0.01	15 +/- 3	13 +/- 3
Galactose wild-type	0.42 +/- 0.004	0.39 +/- 0.004	0.21 +/- 0.01	100 +/- 6	86 +/- 9
Galactose $\Delta galS$	0.41 +/- 0.01	0.44 +/- 0.01	0.42 +/- 0.01	4 +/- 1	-3 +/- 2

[§] In order to calculate an error bar for the predicted O-curve maximum, we fitted the model to each of three day-day repeats and then took the mean and s.e.m. of these three fits.

Figure S8. Model quantitatively predicts the O-curve of the engineered systems. O-curve data for engineered and native sorbitol (a) and galactose (b) systems from Figure 4 are shown with the best fit models. The models for the native and engineered systems have equal values for all shared parameters (see Supplemental Text C). Error bars are standard errors of the mean from 3 day-day repeats. Quality of fit for sorbitol: $R^2 = 0.74$, $p\text{-val} = 8.1e-5$, $RMSE = 0.07/h$, $EV = 74\%$. Quality of fit for galactose: $R^2 = 0.93$, $p\text{-val} = 1.4e-9$, $RMSE = 0.05/h$, $EV = 92\%$. Green square: measured O-curve maximum, blue square: predicted O-curve maximum, red circle: endogenous control point.

Reviewer #4 (Remarks to the Author):

I was asked to review this manuscript, with special attention to evaluating whether the responses to questions and issues raised by former Reviewer #3 were appropriately addressed in the revised manuscript.

In addition to corroborating the overall impression that this constitutes an interesting and original paper, I found that the authors adequately addressed all the issues raised by Reviewer #3.

We thank this reviewer for this endorsement.

One very minor point is that the authors mention the explicit carbon concentrations in the rebuttal letter (4mM (maltotriose) - 27mM (glycerol)), but I could not find explicit mention of these values in the manuscript. I would suggest to add the values, as requested by the reviewer.

We added the explicit molar concentrations as suggested to the Methods section. On pg. 13 we now write:

“The concentration of all carbon sources was 0.2% w/v, except glycerol which was 0.2% v/v and pyruvate in Figure S10a, which was 0.1% w/v. The nutrient concentrations ensure saturation of nitrogen and carbon⁵¹, ranging between 4mM (maltotriose) and 27mM (glycerol).”

Reviewers' Comments:

Reviewer #2 (Remarks to the Author)

The authors have satisfactorily addressed all of our concerns.

REVIEWERS' COMMENTS:

Reviewer #2 (Remarks to the Author):

The authors have satisfactorily addressed all of our concerns.

We thank this reviewer for this endorsement.